# Genetic Diversity and Population Genetic Structure of *Aedes albopictus* in the Yangtze River Basin, China

**DOI:** 10.3390/genes13111950

**Published:** 2022-10-26

**Authors:** Heng-Duan Zhang, Jian Gao, Chun-Xiao Li, Zu Ma, Yuan Liu, Ge Wang, Qing Liu, Dan Xing, Xiao-Xia Guo, Teng Zhao, Yu-Ting Jiang, Yan-De Dong, Tong-Yan Zhao

**Affiliations:** State Key Laboratory of Pathogen and Biosecurity, Beijing Institute of Microbiology and Epidemiology, Beijing 100071, China

**Keywords:** *Aedes albopictus*, genetic diversity, haplotype, microsatellite loci, population structure

## Abstract

*Aedes albopictus* is an indigenous primary vector of dengue and Zika viruses in China. Understanding the population spatial genetic structure, migration, and gene flow of vector species is critical to effectively preventing and controlling vector-borne diseases. The genetic variation and population structure of *Ae. albopictus* populations collected from 22 cities along the Yangtze River Basin were investigated with nine microsatellite loci and the mitochondrial *CoxI* gene. The polymorphic information content (PIC) values ranged from 0.534 to 0.871. The observed number of alleles (*Na*) values ranged from 5.455 to 11.455, and the effective number of alleles (*Ne*) values ranged from 3.106 to 4.041. The Shannon Index (I) ranged from 1.209 to 1.639. The observed heterozygosity (*Ho*) values ranged from 0.487 to 0.545. The *F_IS_* value ranged from 0.047 to 0.212. All *Ae. albopictus* populations were adequately allocated to three clades with significant genetic differences. Haplotype 2 is the most primitive molecular type and forms 26 other haplotypes after one or more site mutations. The rapid expansion of high-speed rail, aircraft routes and highways along the Yangtze River Basin have accelerated the dispersal and communication of mosquitoes, which appears to have contributed to inhibited population differentiation and promoted genetic diversity among *Ae. albopictus* populations.

## 1. Introduction

*Ae. albopictus* is an important vector of dengue fever, chikungunya fever, and Zika fever globally [1] and is one of 100 of the World’s Worst Invasive Alien Species (Global Invasive Species Database (2019). Downloaded from http://www.iucngisd.org/gisd/100_worst.php (accessed on 12 December 2000)). Except for Antarctica, the distribution area of *Ae. albopictus* expands worldwide [2,3,4]. *Ae. albopictus* is a native mosquito species in Asia, including in most parts of China, where the distribution range of the species includes southern Chaoyang, Liaoning Province, eastern Baoji, Shanxi Province and Tianshui, Gansu Province [5]. Compared with *Ae. aegypti*, which is found only in several provinces in the southern part of China, *Ae. albopictus* is the main vector of dengue fever in mainland areas and the main blood-feeding mosquito in urban and rural areas of China [6].

In China, the shipping capacity of the Yangtze River exceeds 71,000 km per year, which is 56% of the total inland river. Meanwhile, its passenger capacity and the volume of freight traffic are 180 million people and 4.714 billon tons, respectively, which is 65% of the total passenger capacity of water passenger volume and 70% of the total water freight traffic volume, respectively. Due to its massive transportation capacity, a comprehensive transportation system of waterway networks, on-land traffic, railways and aviation is well developed alongside the Yangtze River. Mosquitoes can disperse through man-aided transport, especially species of Aedes, involving diapause or desiccative eggs and larvae in container water, and expansion of *Ae. aegypti*, *Ae. albopictus*, *Ae. japonicus* and *Ae. koreicus* beyond its original distribution to local, regional and intercontinental areas occurs via river boats, sea vessels, automobiles and airplanes [7,8,9,10,11,12,13,14,15,16,17,18,19,20,21]. Research to date has investigated the relationship between transportation and the population genetics of mosquitoes, and population genetic studies in Eastern Argentina indicate that passive dispersal of *Ae. aegypti* is supported by a low diversity of haplotypes [22]. Population genetic analysis based on the seven microsatellite loci shows that *Ae. aegypti* is relatively differentiated among 15 seaports in the Philippine Archipelago: low genetic structure and considerable gene flow was found in busy ports, with even distant ports being genetically similar; in small ports, even nearby ports, marine human-mediated long-range dispersal plays a pivotal role in determining the population structure of *Ae. aegypti* [23]. It has also been suggested that dispersal of *Ae. aegypti* is more affected by cargo shipments than by passenger ships. This indicates that *Ae. aegypti* gene flow among sub-populations is greatest between locations with heavy marine traffic [24].

Dispersal patterns and population genetic structure have been analysed to reveal the genetic diversity and gene flow of populations of vector mosquitoes and to supply population genetic information for surveillance as well as for assessing vector competence and integrated management of the species. A series of studies on the population genetics of *Ae. albopictus* probes showed worldwide dispersal and invasion patterns at regional and country levels based on microsatellites and SNPs in the genome and mitogenome to reveal population diversity [6,25,26,27,28]. The results showed that the low differentiation and high co-ancestry identified among *Ae. albopictus* populations in the native range, including China, Thailand and Japan, indicates the maintenance of high genetic connectivity of these populations. Using 13 nuclear microsatellite loci and mitochondrial *CoxI* sequences, it was determined that *Ae. albopictus* was most likely introduced to New Guinea via mainland Southeast Asia before colonizing the Solomon Islands via either Papua New Guinea or SE Asia, which supports that the recent incursion into northern Australia’s Torres Strait Islands was seeded chiefly from Indonesia. This study was the first to document evidence of a recently colonised population (the Torres Strait Islands) that has undergone rapid temporal changes in its genetic makeup resulting from genetic drift or representing a secondary invasion [29].

Female *Ae. albopictus* individuals can fly a limited distance of 500 m for blood feeding and oviposition, and long-distance dispersal of *Ae. albopictus* may occur through the convenient transportation system and the broad international, national and regional trade of tires, lucky bamboo and other items, where larvae and eggs could survive in container water [30]. The present research uses microsatellites and a mitochondrial locus to focus on the population genetic makeup of populations and the genetic diversity of *Ae. albopictus* in cities and counties along the Yangtze River Basin to reveal its dispersal and population genetic structure under conditions of convenient and heavy transport and to aid in the surveillance and control of the species.

## 2. Materials and Methods

### 2.1. Mosquito Sampling

*Ae. albopictus* individuals were collected between July and September 2018 at 22 cities along the Yangtze River Basin, China (Figure 1, Table 1). Larvae were surveyed using the container method; small, indoor, and outdoor water receptacles were examined for larvae at each sampling site. Larvae were collected in the field with a straw and raised in the laboratory. To avoid inbreeding interference, each pool of larval mosquitoes for a given locality was collected from at least five wild breeding places within 500 m. For each of the samples, the larvae were reared separately and emerged at 25 ± 1 °C at 75 ± 5% relative humidity under 14 h light/10 h dark photoperiod, and then adults were identified as *Ae. albopictus* after emergence.

### 2.2. Genotyping of Mosquito Samples

A total of 624 mosquitoes from each of the 22 sample sites were genotyped, and at most 3 to 5 adult mosquitoes were picked out from the same wild breeding place. DNA of individual mosquitoes was extracted following the steps provided with a TAKARA DNA kit and stored at −30 °C until analysis. Individuals were genotyped based on variation at nine microsatellite loci [31]. Approximately 1~2 μL of PCR product was diluted with 10 μL of autoclaved distilled water for DNA. Two microlitres of the diluted PCR product was added to 7.75 μL Hi DiTM formamide and 0.25 Gene Scan-500 LIZTM. The mixture was heated at 94 °C for 5 min and then immediately chilled on ice for 2 min. Genotyping was carried out using a Genetic Analyser3130 ×l (A Life Technologies, Carlsbad, CA, USA). Following Bonacum J. [32], *CoxI* sequence polymorphisms among the 624 individuals were investigated. Briefly, DNA amplification of a 550-bp fragment of *CoxI* was performed using a T100 Thermal Cycler with a primer set (5′-GGAGGATTTGGAAATTGATTAGTTC-3′(F-*CoxI*) and 5′-CCCGGTAAAATT AAAATATAAACTTC-3′ (R-*CoxI*)) in a 50 µL reaction mixture containing 10 µL PCR buffer, 4 µL dNTPs, 1 µL primers, 0.5 µL Prime STAR HS DNA polymerase, and 34.5 µL ddH_2_0. The PCR amplification program was set as follows: pre-denaturation at 94 °C for 3 min, followed by 35 cycles of denaturation at 94 °C for 30 s, annealing at 54 °C for 45 s and elongation at 72 °C for 1 min, with a final extension at 72 °C for 10 min. All PCR products were separated by 2% agarose gel electrophoresis for detection. The target fragments for *CoxI* were excised from the gel under UV light and purified with GenElute™ PCR Clean-Up Kit. Each purified PCR product was cloned into the PCR™2.1 vector using TA Cloning™ Kit and selected by bacterial liquid PCR using T7 promoter primers. Finally, at least 20 positive clones for each PCR product were sequenced on both strands using an ABI 3730 × L automatic sequencer (Life Technologies, Carlsbad, CA, USA).

### 2.3. Data Analysis

Polymorphic information content (PIC) for each locus was calculated using POPGENE Version 1.3 and PIC-Calc 0.6. Genetic diversity (expected (*He*), observed (*Ho*) heterozygosity), the mean number of alleles (*Na*), and F-statistic parameters (F_IS_, F_ST,_ and F_IT_) were estimated from allele frequencies with FSTAT 2.9.3.2 [33]. For each locus–population combination for the global data set and population groupings, Fisher’s exact test with Bonferroni correction was employed to test for possible deviations from the Hardy–Weinberg equilibrium (*HWE*) using GENEPOP’007 [34]. Exact *p* values were calculated using the Markov chain algorithm with 10,000 dememorizations, 500 batches, and 5000 iterations per batch. Arlequin software 3.5.2.2 was used to assess pairwise differences between populations [35] and perform AMOVA [36] based on allelic frequency. In addition, the F_ST_ of each population was calculated from the mean value of pairwise differences between populations. An excess of heterozygosity in a significant number of loci (calculated via Bottleneck 1.2.02) is an indicator of a genetic bottleneck, whereas the opposite (i.e., He, Heq) [37] suggests population expansion. NTsys 2.01 software was employed to build the UGPMA tree and the DAPC analysis was performed on the microsatellite data of each population with the R packages “Adegenet 2.1.3”. IBD analysis was also calculated via the R packages “Adegenet 2.1.3” with the comparation between genetic distance and geographic distance of all populations. The haplotypes of all populations were screened with DnaSP version 6.0 [38], and genetic relationships among all haplotypes were displayed as a TCS network constructed by Network 10.0.0.0 [39,40]. Moreover, Tajima’s *D* test was tested via Arlequin 3.5.2.2 to determine the population diffusion of each population, while dispersal routes among 7 locations were calculated via the R packages “divMigrate” with the number of bootstrap replicates set to 3, the α value set to 0.05, the Nm method chosen for the migration statistic and the filter threshold value set to 0.15.

## 3. Results

### 3.1. Genetic Diversity and Variation

In the present study, 22 *Ae. albopictus* populations were collected along the Yangtze River Basin (Table 1 and Figure 1). Nine pairs of microsatellite markers were found to be highly polymorphic and were therefore chosen for microsatellite genotyping analysis (Appendix A). The PIC values of each locus range from 0.534 to 0.871. According to the definition of PIC values provided by Allah et al. [41], all of the selected markers are highly informative (PIC value > 0.5) (Appendix A). As shown in Table 2, the observed number of alleles (Na) and the effective number of alleles (Ne) of each population range from 5.455~11.455 and from 3.106~4.041, respectively. The mean Na value of the midstream and downstream Yangtze River Basin populations (10.955) was calculated to be higher than that of the upstream Yangtze River Basin populations (8.364), except for Shanghai (5.455). Similarly, except for Shanghai (3.106), the mean Ne of the midstream and downstream populations (4.082) is higher than that of the upstream populations (3.780). The Shannon index (I) of each population ranges from 1.209 to 1.639. The mean I value of the midstream and downstream Yangtze River Basin populations (1.588) is higher than that of the upstream Yangtze River Basin populations (1.538); the I in Hubei is highest (1.639) and that in Shanghai lowest (1.209). The observed heterozygosity (Ho) values of all areas range from 0.487 to 0.545, which are lower than the expected heterozygosity (He) values (ranging from 0.617 to 0.709). Except for Shanghai, the F_IS_ value of each population ranges from 0.047 to 0.212, and each *Ae. albopictus* population exhibited significant departure from HWE.

### 3.2. Population Structure and Differentiation Based on Microsatellite Analysis

In population structure analysis, all *Ae. albopictus* populations were adequately allocated to three groups with *k*-means calculation function of Adegenet 2.1.3 and the geographical distribution of samples (Figure 2a). Group I includes Shanghai, which was completely separate from the other six regions; Group II includes Hubei, Chongqing and Sichuan; and Group III includes Anhui, Jiangsu and Jiangxi. DAPC analysis showed the same grouping results (Figure 2b). AMOVA results are presented in Appendix A. From the total genetic variation partitioned, 2.03% could be attributed to differences among individuals within populations and 97.97% to differences within individuals (F_ST_ = 0.0203, F_IS_ = 0.17846, F_IT_ = 0.19514, and all *p* < 0.00001). In addition, pairwise F_ST_ values between populations range from 0.002 (Anhui and Jiangsu) to 0.108 (Shanghai and Chongqing). Overall, genetic differentiation between *Ae. albopictus* populations in Shanghai and the other six regions was found to be moderate, whereas the genetic differentiation levels among the other six regions is negligible (Table 3). We detected frequent communication among *Ae. albopictus* populations in the Yangtze River Basin, with the population diffusion path mainly spreading from Chongqing (CQ), Shanghai (SH), Jiangsu (JS), Jiangxi (JX) and Anhui (AH) to Hubei (HB), where gene flow (Nm) among populations was observed to be greater than 0.70, except in Shanghai (Figure 3). Meanwhile, a significant bottleneck effect was also detected among *Ae. albopictus* populations from all seven locations (Table 4). In contrast to the high individual variation observed, no significant positive correlation among all individuals was detected by isolation by distance (IBD) analysis (*p* = 0.147); While significant positive correlation (*p* = 0.009) between the genetic distance and the geographical distance of all individuals, when the Shanghai population was eliminated (Appendix A).

### 3.3. Haplotype Diversity and Network Analysis Based on CoxI Sequences

A total of 27 CoxI haplotypes were found among 624 *Ae. albopictus* individuals based on analysis of CoxI sequences, and the haplotype indices changed dramatically across *Ae. albopictus* populations from different areas (Table 5). Across all populations, the number of polymorphic sites ranges from 1 (Shanghai) to 13 (Hubei). Except for Shanghai (Hd = 0.000; π = 0.000), the haplotype diversity (Hd) ranges from 0.074 (Chongqing) to 0.393 (Hubei), with nucleotide diversity (π) ranging from 0.0002 (Chongqing) to 0.001 (Anhui and Hubei). As illustrated in Appendix A, a TCS network was reconstructed with all 624 CoxI sequences, and the results showed haplotype 2 (H2) to be the most common molecular type and the most frequent haplotype in *Ae. albopictus* populations in different regions of the Yangtze River Basin. Additionally, H2 forms 26 other haplotypes after one or more site mutations.

## 4. Discussion

The number of alleles, heterozygosity, and polymorphism information content (PIC) are common indices to evaluate the genetic diversity of populations [42,43,44], and PIC is a relatively good index of gene fragment polymorphism: the higher the population heterozygosity, the greater the genetic diversity and levels of polymorphism and the greater the potential variation for natural selection to act on. A *He* value between 0.5 and 0.8 reveals high genetic polymorphism in a population [45]. The average expected heterozygosity in this study was found to be 0.677, indicating high polymorphism. The average PIC of 0.706 suggests highly polymorphic loci [6,46,47]. Such variation in the nine microsatellites shows that the 22 *Ae. albopictus* populations are relatively rich in terms of PIC and have relatively even gene frequency distributions in the Yangtze River Basin. This approach is a highly effective and reliable way of assessing genetic diversity, and the results also show that the sample size was more than adequate. Analysis of *CoxI* polymorphism data for *Ae. albopictus* revealed different haplotypes in each population; 27 haplotypes were found in 624 samples, which suggests high genetic diversity.

According to the results of population differentiation analysis, all *Ae. albopictus* populations were adequately allocated to three clades with significant genetic differences. Shanghai was completely separated from the other six regions, and the Fst values > 0.05 in the population of Shanghai and the other six regions indicated a moderate level [43]. Previous studies have shown that the spread of mosquitoes is closely related to human activities [48]. Genome-wide SNPs have been used to reveal drivers of gene flow in Guangzhou, and the results showed that human transportation networks, particularly shipping terminals, influence the genetic structure of *Ae. albopictus* populations [49]. Landscape genetics and information-theoretic model selection have also been used to evaluate the relative roles of human aid and natural movement. In general, highways and water way availability are important and facilitate dispersal at a broad spatial scale [50]. With the rapid development of China, a comprehensive transportation system has formed alongside the Yangtze River Basin, including water transport, railways, highways, and aviation. Based on the result of diffusion analysis that frequent communication was detected among *Ae. albopictus* populations in the Yangtze River Basin and the significant bottleneck effect of each *Ae. albopictus* population, we hypothesised that the rapid expansion of high-speed rail, aircraft routes, and highways along the Yangtze River Basin has not only facilitated trade among different cities but has also accelerated the spread of mosquitoes. Meanwhile, compared with other cities, Shanghai’s disinfection work may be more thorough, resulting in a single diversity of mosquitoes.

As a transportation hub, Hubei Province connects the upper and lower Yangtze River. According to the results of haplotype diversity, Hubei *Ae. albopictus* populations contain 13 polymorphic sites, which is much higher than in other areas. The Shannon index was also richest in Hubei due to the high gene flow (*Nm* > 0.70) between Hubei and the other collection sites, except Shanghai (*Nm* = 0.17). This may be due to natural and geographical factors in Hubei, which is known as the “province of thousands of lakes”. Hubei has a rich ecosystem and various geomorphological types, including mountains, hills, and plains. Hubei has a strong high-speed rail hub network, air transport network, highway network and water transport system, directly connecting it with 10 other provinces and cities in China. *Ae. albopictus* populations can disperse and communicate via convenient transportation [51], which is the main reason for the greater number of haplotypes and richer haplotype diversity in Hubei.

In general, the genetic distance of populations and geographic separation are positively correlated; however, no significant correlation between these two variables was observed among all populations in the present study. Moreover, the genetic distance of *Ae. albopictus* displays no relationship with geographic distance in a study in which the mosquitos were sampled from 19 sites across 2512 km^2^ of volcanic Réunion Island [52], with different results in the Indo-Pacific, where the sample sites included Australasia, SE Asia, the Indian Ocean, Pacific Ocean islands and the United States [30]. This model assumes that populations that are closer to the original population should genetically be much more similar than those that are distant from the original population. At the same time, this satisfies the model’s prerequisite of obvious genetic and geographic distance between populations. Based on AMOVA, the most genetic variation in *Ae. albopictus* was detected within individuals, at 97.97%. Long-distance dispersal of mosquitoes including *Ae. albopictus* and *Ae. aegypti* is driven by human activity, with particular emphasis on human trafficking [14,16,17,18,21,22,23,24,29,48,50,53,54], resulting in high gene flow and low genetic distance along the Yangtze River Basin.

F_IS_ is an index of inbreeding within populations. In present study, all F_IS_ values tested are positive, which means different degrees of inbreeding exist in the populations. Meanwhile, *Ho* values were lower than *He* values, which also indicated high gene consistency for the *Ae. albopictus* populations and that the population might be inbreeding. A combination of anthropogenic and natural factors, such as eradication of breeding habitat, high temperature and decreased rainfall, may lead to breeding habitat fragmentation and reduced population size and therefore increase the likelihood of inbreeding, leading to a heterozygosity deficit [55,56]. Nevertheless, the neutral Tajima’s *D* test results showed that all populations were negative, suggesting that *Ae. albopictus* populations in the Yangtze River Basin are expanding. This may be due to the influence of hosts and various vehicles in the area.

## 5. Conclusions

The present study systematically evaluated the genetic variation, population structure and haplotype-based phylogenetic relationships of *Ae. albopictus* populations in the Yangtze River Basin. All *Ae. albopictus* populations were adequately allocated to three clades with significant genetic differences. Continuous dispersal and communication, as aided by human activities and transportation, appears to contribute to inhibited population differentiation and to promote genetic diversity among *Ae. albopictus* populations in the Yangtze River Basin.

## Figures and Tables

**Figure 1 genes-13-01950-f001:**
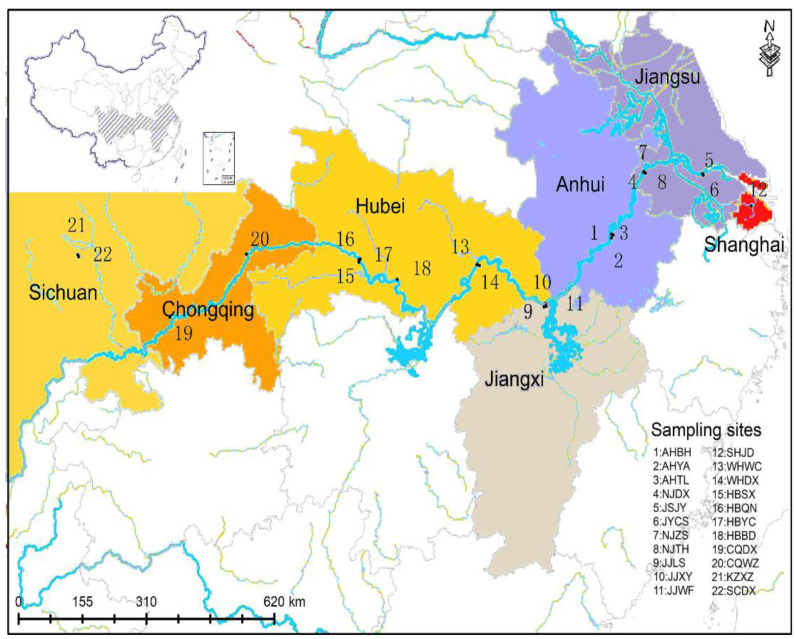
*Ae. albopictus* individuals were collected between July and September 2018 at 22 cities along the Yangtze River Basin.

**Figure 2 genes-13-01950-f002:**
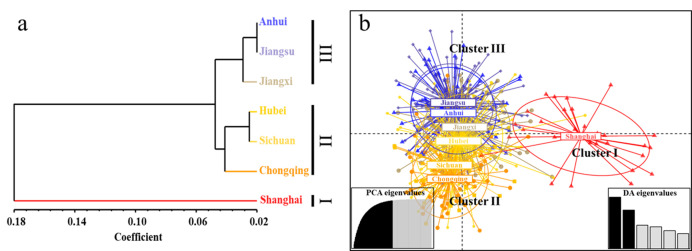
UGPMA (**a**) and DAPC (**b**) analysis of *Ae. albopictus* population structure based on microsatellite loci in the Yangtze River Basin. The *x*-axis for Figure 2a stands for the Dice Coefficient of Similarity. For Figure 2b, eighty first principal components (PCs) and six discriminant eigenvalues were retained during the analyses to describe the relationship between the clusters. The axes represent the first two linear discriminants (LD). Each circle represents a cluster and each colour represents the different subpopulations identified by the DAPC.

**Figure 3 genes-13-01950-f003:**
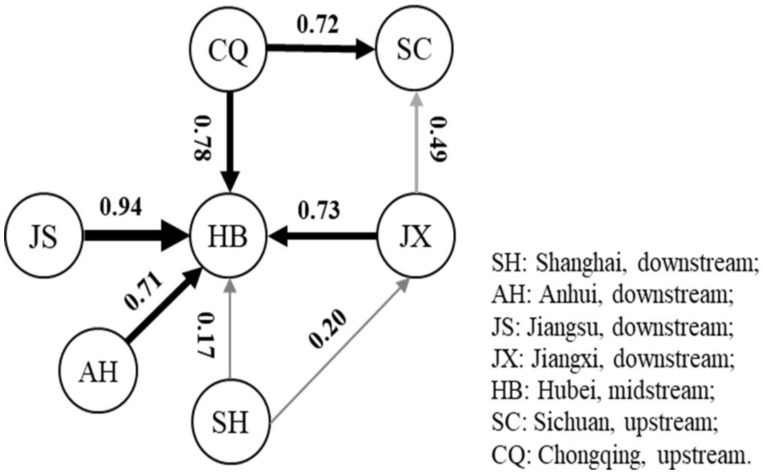
Diffusion analysis of *Ae. albopictus* population in the Yangtze River Basin based on microsatellite loci. Each circle stands for the *Ae. albopictus* population, the values (Nm) stand for the gene flow between two populations and the arrows stand for the direction of the gene flow, which varies with the gene flow value; the diffusion routines with Nm value greater than 0.2 are displayed.

**Table 1 genes-13-01950-t001:** Sampling information of all 22 *Ae. albopictus* populations distributed in various regions of the Yangtze River.

No.	Location	Population Code	Geographic Coordinate	Sample Size
1	**Anhui** (downstream)	AHBH	N30°53′10.41″ E117°46′18.71″	25
2		AHYA	N30°57′31.85″ E117°47′14.68″	30
3		AHTL	N30°56′12.57″ E117°49′16.35″	30
4	**Jiangsu** (downstream)	NJDX	N32°03′25.02″ E118°46′27.42″	26
5		JSJY	N31°54′05.29″ E120°16′20.34″	28
6		JYCS	N31°55′31.69″ E120°15′19.61″	30
7		NJZS	N32°05′19.69″ E118°43′54.56″	30
8		NJTH	N32°04′14.44″ E118°45′54.30″	30
9	**Jiangxi** (downstream)	JJLS	N29°40′07.78″ E115°56′54.98″	28
10		JJXY	N29°40′45.15″ E116°00′17.82″	30
11		JJWF	N29°43′54.65″ E115°59′44.95″	30
12	**Shanghai** (downstream) (downstream)	SHJD	N31°12′03.49″ E121°25′41.03″	25
13	**Hubei** (midstream)	WHWC	N30°33′18.14″ E114°17′35.34″	30
14		WHDX	N30°31′52.51″ E114°21′12.32″	30
15		HBSX	N30°43′50.35″ E111°18′24.41″	26
16		HBQN	N30°43′13.14″ E111°16′16.61″	27
17		HBYC	N30°40′06.23″ E111°16′01.27″	30
18		HBBD	N30°19′33.84″ E112°14′28.09″	30
19	**Chongqing** (upstream)	CQDX	N29°34′01.11″ E106°27′51.77″	27
20		CQWZ	N30°49′17.86″ E108°22′40.18″	30
21	**Sichuan** (upstream)	KZXZ	N30°39′57.73″ E104°03′00.46″	22
22		SCDX	N30°37′57.88″ E104°04′52.49″	30

AHBH: AnHuiBaiHe; AHYA: AnHuiYiAn; AHTL: AnHuiTongLin; NJDX:NanJingDaXue; JSJY: JiangSuJiangYin; JYCS: JiangYinChunShen; NJZS: NanJingZhongShan; NJTH: NanJingTianHe; JJLS: JiuJiangLuShan; JJXY: JiuJiangXueYuan; JJWF: JiuJiangWanFu; SHJD: ShangHaiJiaoDa; WHWC: WuHanWuChang; WHDX: WuHanDaXue; HBSX: HuBeiShanXia; HBQN: HuBeiQiaoNan; HBYC: HuBeiYichang; HBBD: HuBeiBiDian; CQDX: ChongQingDaXue; CQWZ: ChongQingWanZhou; KZXZ: KuanZhaiXiangZi; SCDX: SiChuanDaXue.

**Table 2 genes-13-01950-t002:** Genetic diversity indices of 22 *Ae. albopictus* populations sampled from seven locations of the Yangtze River Basin based on nine microsatellite loci.

Location	n_a_	n_e_	I	Ho	He	WHD	F_IS_
Anhui	11.455 ± 2.770	4.041 ± 2.208	1.610 ± 0.486	0.532 ± 0.228	0.688 ± 0.177	0.252 ***	0.143 ***
Jiangsu	11.273 ± 3.349	3.896 ± 1.841	1.578 ± 0.478	0.487 ± 0.229	0.678 ± 0.187	0.288 ***	0.210 ***
Jiangxi	9.818 ± 3.093	4.166 ± 2.227	1.576 ± 0.498	0.532 ± 0.229	0.692 ± 0.166	0.272 ***	0.150 ***
Shanghai	5.455 ± 2.115	3.106 ± 1.436	1.209 ± 0.469	0.545 ± 0.212	0.617 ± 0.187	0.123	0.047
Hubei	11.273 ± 3.319	4.223 ± 1.831	1.639 ± 0.456	0.513 ± 0.216	0.709 ± 0.161	0.329 ***	0.212 ***
Chongqing	8.455 ± 2.423	3.649 ± 1.634	1.460 ± 0.466	0.495 ± 0.244	0.659 ± 0.196	0.313 ***	0.171 ***
Sichuan	8.273 ± 1.849	3.910 ± 1.628	1.514 ± 0.430	0.522 ± 0.218	0.693 ± 0.176	0.342 **	0.168 ***

“± values” are standard deviations. na means numbers of alleles; ne means numbers of effective alleles; I means Shannon index; Ho means observed heterozygosity; He means expected heterozygosity; WHD means Hardy–Weinberg disequilibrium; FIS means inbreeding coefficient. ** means *p*-value < 0.01, *** means *p*-value < 0.001.

**Table 3 genes-13-01950-t003:** Genetic difference (F_ST_) among 22 *Ae. albopictus* populations sampled from seven locations of the Yangtze River Basin.

	Anhui	Jiangsu	Jiangxi	Hubei	Chongqing	Sichuan	Shanghai
Anhui		0.155 ± 0.004	0.004 ± 0.001	0.000 ± 0.000	0.000 ± 0.000	0.000 ± 0.000	0.000 ± 0.000
Jiangsu	0.002		0.000 ± 0.000	0.000 ± 0.000	0.000 ± 0.000	0.000 ± 0.000	0.000 ± 0.000
Jiangxi	0.008	0.013		0.000 ± 0.000	0.000 ± 0.000	0.000 ± 0.000	0.000 ± 0.000
Hubei	0.011	0.011	0.015		0.000 ± 0.000	0.002 ± 0.000	0.000 ± 0.000
Chongqing	0.022	0.027	0.024	0.015		0.000 ± 0.000	0.000 ± 0.000
Sichuan	0.018	0.021	0.019	0.009	0.015		0.000 ± 0.000
Shanghai	0.073	0.075	0.067	0.067	0.108	0.082	

The numbers below the diagonal are F_ST_ values among each *Ae. albopictus* population, while numbers above the diagonal are *p* values for each F_ST_ value. The ± values are standard deviation.

**Table 4 genes-13-01950-t004:** Heterozygosity tests of 22 *Ae. albopictus* populations among seven locations based on the stepwise mutation model (S.M.M.) and two-phase model (T.P.M.).

Yangtze River Basin		Anhui	Jiangsu	Jiangxi	Hubei	Chongqing	Sichuan	Shanghai
S.M.M.	He < Heq *	11	11	10	11	10	10	8
	He > Heq *	0	0	1	0	1	1	3
	*p* (He < Heq)	0.000	0.000	0.001	0.000	0.001	0.001	0.032
T.P.M.	He < Heq	9	9	6	8	7	6	4
	He > Heq	2	2	5	3	4	5	7
	*p* (He < Heq)	0.007	0.007	0.265	0.031	0.111	0.265	0.514

*: He < Heq shows the numbers of loci showing a heterozygote deficient, while He > Heq shows the number of loci showing a heterozygote excess; *p* (He < Heq): *p*-value of Wilcoxon tests to determine the significance of the number of loci in which He < Heq.

**Table 5 genes-13-01950-t005:** The genetic diversity indices of all 22 *Ae. albopictus* populations sampled from seven locations of the Yangtze River Basin via *CoxI* analysis.

NO.	Location	Region	H	Hd	π	k	Tajima’s *D*	Fu’s Fs	Haplotypes
1	downstream	Anhui	9	0.315	0.001	0.341	−1.976 *	−1.624	H1(3),H2(67),H3(2),H4(2),H5(1),H6(1),H7(2),H8(2),H9(1)
2	downstream	Jiangsu	6	0.179	0.0004	0.185	−1.669	−2.407 *	H2(105),H3(1),H15(1),H16(6),H23(2),H24(1)
3	downstream	Jiangxi	7	0.369	0.0008	0.403	−1.603	−1.154	H2(57),H9(4),H10(1),H15(2),H17(5),H20(2),H21(1),
4	downstream	Shanghai	1	0.000	0.000	0.000	0.000	0.000	H2(25)
5	midstream	Hubei	13	0.393	0.001	0.436	−1.958 *	−2.362 *	H2(135),H4(2),H9(2),H10(4),H11(3),H12(1),H15(9),H16(1),H17(9),H18(5),H19(1),H21(1)
6	upstream	Chongqing	4	0.074	0.0002	0.075	−1.639	−3.251 **	H2(77),H12(1),H13(1),H14(1)
7	upstream	Sichuan	7	0.315	0.0007	0.337	−1.885 *	−3.229 *	H2(43),H3(1),H9(1),H15(1),H22(1),H25(4),H26(1)

H means the types of haplotypes; Hd means haplotype diversity; π means nucleotide diversity; k means the average number of nucleotide differences. * means *p*-value < 0.05; ** means *p*-value < 0.01.

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
