# Peer review of "Genetic Diversity and Population Genetic Structure of Aedes albopictus in the Yangtze River Basin, China"

_genes, 2022, doi:10.3390/genes13111950_

Round 1

Reviewer 1 Report

Pretty good study design, but a number of problems in the display and interpretation of results that must be addressed before publication. I don't think this should be too difficult for the authors, I hope.

Major concerns (all must be addressed before publication):

Line 140: The authors never explain how they divided their sites into their 7 populations, but it looks like it was just by political boundaries, which the mosquitoes obviously do not care about. These populations seem quite questionable to me: for instance, Jiangxi has 3 sites within kilometers of each other, while Hubei has 6 sites separated by 300 km, and two of those sites are closer to all three Jiangxi sites than to any of the other Hubei sites. I just don't see any reason to believe that these are sensible sampling units. I recommend that the others use an individual-based method for describing genetic diversity, like PCA or STRUCTURE, to look at all the individuals at once and then use that determine a posteriori what their populations should be. Maybe PCA/STRUCTURE will show that there is substantial albo population structure divided up by province, or that there is a smooth cline, or no population structure, in which case the reader will be assured that the current population structure is appropriate. Otherwise, the results will guide the authors into making more sensible populations.

190-191: You need to explain how you came to the conclusion that there are 3 population clusters, because I don't get it. You didn't seem to do any quantitative analysis (like the Evanno method or a likelihood analysis in DAPC) that could tell you which cluster number best fits your data, and qualitatively I do not see how Figure 2 shows that there are 3 clusters in your data. 2a can be interpreted as "3 clusters", sure, but you can just as well use it as evidence for 2 clusters, or 4, 5, 6, or 7 clusters. And on 2b the two clusters look like they form a single continuous mass to me. You can probably make the argument for 3 clusters here, but you need to make it (and ideally actually do a quantitative analysis). Also, your 3 clusters are very much undermined by the fact that you suddenly switch from this 3-cluster set (Hubei-and-upriver, downriver-from-Hubei, Shanghai) to a completely different 3-cluster set in Figure 5 (upriver-from-Hubei, Hubei, and downriver-from-Hubei-including-Shanghai), without informing the reader or explaining why this is the case. You need to explain why you switch clustering methods or, better, remain consistent across analyses.

Figures and tables: Figures and tables are currently not acceptible for publication. All of the figures are far too pixelated to read, and none of the figures or tables have a caption that adequately explains it. I provide some specific feedback below:

Figure 1: The inset has a caption that is completely unreadable, and the box with the sampling sites is almost entirely unreadable. The color scheme of the 7 populations does not match the color scheme of figure 2. When printed out, the small gray rivers are hard to see against the background of Sichuan province, making it look like 21 and 22 are not on a river, which raised questions for me.

Figure 2: You never say which facet shows the results of UPGMA and which shows DAPC. What is the x-axis for figure 2A? How should we interpret it? What are the axes for figure 2b? What do the points represent? What are some points different shapes from others? Why are some points different colors from others? What are the ovals and lines? What are PCA eigenvalues and what does this facet mean? What are DA eigenvalues and what does this facet mean? What should I take away from this figure?

Figure 3: What are these numbers and how should I interpret them? Arrows vary in color and thickness, what does that mean? (As an aside, I THINK that you looked at every possible pair of populations and only present the results that are significant? If I am right, you need to explain this in both the caption and the text. If I am wrong, you need to explain somewhere how you chose these 8 comparisons to examine. Also, you may wish to explain how this figure accounts for the possibility of migration in both directions--i.e., if there is equal migration both ways, what would the result be? Does the 0.2 arrow mean that there is little migration b/w SH and JX, or could it mean that there is a lot of migration in both directions, but slightly more from SH->JX than from JX->SH?)

Figure 4: I recommend deleting this figure or sending it to the SI; it doesn't really give you any more value than just stating the p-value for this analysis. If you do insist on keeping the figure, please explain what the axes are and what the little red line is.

Figure 5: What do the size of the circles mean (I think this is on the legend, but the legend is unreadable)? What are the notches? 

Table 2: Probably you shouldn't have more than 3 significant figures here, given your sample design. Also, what are the +/- numbers? Standard error? Standard deviations? Please include a caption that briefly describes what every variable here is and what the asterisks mean.

Table 3: I recommend sending this to the SI.

Table 4: I have no idea what is going on with this table; I therefore can't really say what needs to be put in the caption, but something does.

Table 5: Why are there different numbers above and below the diagonal? What are the +/- values?

Table 6: Most of the comments from Table 2 apply here as well.

In general, good captions mean that someone can read your abstract on only the figures, tables, and their captions, and have a fairly good idea what you did in the study. That should be your goal. If I wasn't pretty familiar with almost all of the software you used, I would be extremely confused about what these results were even if I had a strong population genetic background.

Moderate concerns (most should be addressed before publication):

Lines 133-139: I think you need to explain the purpose of this approach. Why did you take CoxI PCR product, clone it into plasmids, then sequence plasmids? Why not just sequence the PCR product directly and save several steps that could introduce spurious genetic diversity? Also, why sequence 20 clones per PCR product? Shouldn't they have the same sequence? What did you do if they didn't? Or did you pool them and only sequence the pool once?

152-154: Authors mention STRUCTURE but do not mention which admixture model they used and never present any results. Delete this passage or improve it and provide results.

157-158: The authors fail to mention what method they used for dividing their data into clusters. Did they use the a posteriori algorithm implemented in DAPC? Did they have an apriori assignment to some number of clusters? Figure 2b is of no help here because the data are divided into 7 clusters by color but 3 clusters are labelled by the authors.

158-160: How did you determine geographic distances between populations in the instances when a population had multiple sites?

190-191: Throughout the results section, results are described without making it clear what analysis produced those results. This needs to be corrected. For instance, you should mention here that this is the result of the UPGMA analysis; other examples may be found later on.

214-216: Throughout the results section, the authors say things like "Except for Shanghai, we found XYZ." In these cases, please also tell us how Shanghai is different.

283-289: Your results certainly do not confirm this. Your results point to migration rates at a single point in time; you do not know whether spread of mosquitoes has accelerated with only one time point. I think that your hypothesis is plausible, but you need to make clear that this is a hypothesis that was produced from your data, not one that was tested or confirmed by your data (and make the appropriate fixes in the abstract as well).

Minor concerns (I think this would make the paper better; authors may ignore):

Lines 40-45: No citation?

46-53: I found this passage hard to understand; check the style?

69-94: This is a very long passage about previous work; it's not to have some amount of this, but this is maybe a bit more than I need since you never almost compare your results to other studies.

95-99: No citation? Also, I think you mean "tires".

173: No citation for Allah et al.

207-209: Interesting possible followups on the IBD that I think might be easy to do:

-Redo this with more sensible populations (see above; possibly ignoring populations altogether and just analyzing it at the site level).

-If you remove Shanghai, is there IBD in the remaining dataset? It seems possible that the large genetic distance between Shanghai and everything else could be swamping a signal of IBD among the remaining sites.

-Site 13 is about equal in distance from 18 and 10. BUT, if you look at the river, it is quite a bit more sailing distance between 13 and 18 than between 13 and 10. What if you made 3 distance matrices: one the distance-as-the-crow-flies that you have now, a second that is the distance along the river between each site, and a third that is the distance along major highways between each site. Then, if there was IBD using, say, highway-distance but not the other two, that would be an important clue that albo disperses more by truck than by river or aerially/by-plane.

291: If the genetic distinctness of Shanghai is driven by inflow of mosquitoes from outside China, wouldn't we expect the Shanghai population to be more diverse than the others, not less diverse?

306-309: Doesn't the first half of this sentence contradict the second?

322-323: I am not aware that any positive value of FIS points to severe inbreeding; could you provide a citation for this please?

Reviewer 2 Report

In general, the study was well conducted, and the methodology sounds perfect but the presentation of the results is not intelligible. Authors need to summarize their results and reduce the number of figures and tables in the manuscript. The finding that the populations of Ae albopictus along the Yangtze river can be divided into 3 groups is not new. So, further analysis within each group should be performed to explore the features of these groups better. Besides, the authors state that Ae albopictus in Shanghai have distinct genetic patterns from other regions. This finding was poorly explored in the text, authors must be explicit about the key parameters that differentiate Ae albopictus in Shanghai from other regions. They also need to discuss the importance of these parameters to the structure of the Ae albopictus population in Shanghai.

Minor concerns:

Lines 24-25: Polymorphic information content (PIC), the first time you mention it. Always define any abbreviation when it is said the first time.

Lines 57-60: Please check all references. Reference 20 is not the corrected study mentioned in this sentence.

Line 65: This reference is also incorrect.

Line 105 (2.1 Mosquito sampling): You should describe the methodology used to classify the larvae of Ae albopictus.

Line 167 (Results): Results were not clearly presented. Reduce the number of tables and figures. Some figures and some tables are redundant. For example figure 5 and table 6. Another example is Figures 2 a and B, they show the same results. In general, tables were poorly explained and poorly explored as well. All figures have low quality.

Line 170: Move table 1 to supplementary material.

Line 209: Figure 4 can also be moved to the supplementary material

Line 221 (3.4. Figures, Tables and Schemes): figures and tables should be presented after they were mentioned in the text and not as a particular separated topic.

Line 226 (Figure 2): Explain PCA and DA eigenvalues in the panel B of this figure

Line 236 (Table 1): Define all abbreviations used in column 3 of this table.

Line 271: What do you mean by separated?? How does the structure of the Ae albopictus population in Shanghai differs from other regions? Try to be more specific about these features of Ae albopictus in Shanghai that make it distinct from insects of other cities.

Lines 279-283: Reformulate this to explain what you want to say.

Lines 283-287: You need to include some references to support these statements

Reviewer 3 Report

The mosquito Aedes albopictus is a native species in Asia, including China, and an important vector of Dengue fever and other serious human diseases. Authors here studied the genetic diversity and the population genetic structure of the mosquitos in the Yangtse river basin using nine microsatellite loci and the mitochondrial Cox I gene. Their results clearly show that all populations from 22 sample sites were allocated to three clades with significant genetic differences. The results also indicate a continuous import of the mosquitoes with ship and airplane from overseas in Shanghai. The rapid expansion of high-speed rail, aircraft routes and highways along the Yangtze River Basin accelerated the spread of mosquitoes through the river basin with the greatest number of haplotypes and richer haplotype diversity around the central city of Hubei.

The study is carefully done, and the results support the thesis of human transport of the mosquitoes. The manuscript needs only minor improvement:

- line 34: haplotype in lowercase letters

- line 54 and others: please give all species names in italics, also in Tables and figure legends and in the References

- line 112; insert a space between number and unit

- line 148: insert a comma in numbers greater than 1,000

- Table 3: "Within" in uppercase letters

- Tables 4 and 6: explain the number of asterisks used in the table legends

- References: use a uniform style of writing in the References (paper titles in lowercase letters, journal titles as abbreviations etc.)

Round 2

Reviewer 2 Report

All comments were fully addressed by the authors. The current version of the manuscript is edible for publication.